# From Trojan Horses to Castle Walls: Unveiling Bilateral Backdoor Effects in Diffusion Models

## Abstract

While state-of-the-art diffusion models (DMs) excel in image generation, concerns regarding their security persist. Earlier research highlighted DMs' vulnerability to backdoor attacks, but these studies placed stricter requirements than conventional methods like 'BadNets' in image classification. This is because the former necessitates modifications to the diffusion sampling and training procedures. Unlike the prior work, we investigate whether generating backdoor attacks in DMs can be as simple as BadNets, *i.e.*, by only contaminating the training dataset without tampering the original diffusion process. In this more realistic backdoor setting, we uncover *bilateral backdoor effects* that not only serve an *adversarial* purpose (compromising the functionality of DMs) but also offer a *defensive* advantage (which can be leveraged for backdoor defense). Specifically, we find that a BadNets-like backdoor attack remains effective in DMs for producing incorrect images that do not align with the intended text conditions and for yielding incorrect predictions when DMs are employed as classifiers. Meanwhile, backdoored DMs exhibit an increased ratio of backdoor triggers, a phenomenon we refer to as 'trigger amplification', among the generated images. We show that this latter insight can enhance the detection of backdoor-poisoned training data. Even under a low backdoor poisoning ratio, we find that studying the backdoor effects of DMs can be valuable for designing anti-backdoor image classifiers. Last but not least, we establish a meaningful linkage between backdoor attacks and the phenomenon of data replications by exploring DMs' inherent data memorization tendencies.

## 1 Introduction

Backdoor attacks have been studied in the context of *image classification*, encompassing various aspects such as attack generation (Gu et al., 2017; Chen et al., 2017), backdoor detection (Wang et al., 2020; Chen et al., 2022a), and reverse engineering of backdoor triggers (Wang et al., 2019; Liu et al., 2019). *In this work, we focus on backdoor attacks targeting diffusion models (DMs)*, state-of-the-art generative modeling techniques that have gained popularity in various computer vision tasks (Ho et al., 2020), especially in the context of text-to-image generation (Rombach et al., 2022).

In the context of DMs, the study of backdoor poisoning attacks has been conducted in recent works (Chou et al., 2023a; Chen et al., 2023a; Chou et al., 2023b; Zhai et al., 2023; Struppek et al., 2022; Huang et al., 2023a). We direct readers to Sec. 2 for detailed reviews of these works. Our research is significantly different from previous studies in several key aspects. ❶ (Attack perspective, termed as '**Trojan Horses**') Previous research primarily approached the issue of backdoor attacks in DMs by focusing on attack generation, specifically addressing the question of whether a DM can be compromised using backdoor attacks. Prior studies impose *impractical* backdoor conditions in DM training, involving manipulations to the diffusion noise distribution, the diffusion training objective, and the sampling process. Some of these conditions clearly violate the 'stealthiness' of backdoor attacks during the training process. Instead, classic BadNets-like backdoor attacks (Gu et al., 2017) only require poisoning the training set without changes to the model training procedure. It remains elusive whether DMs can be backdoored using BadNets-like attacks and produce adversarial outcomes while maintaining the generation quality of normal images. ❷ (Defense perspective, termed as '**Castle Walls**') Except a series of works focusing on backdoor data purification (May et al., 2023;

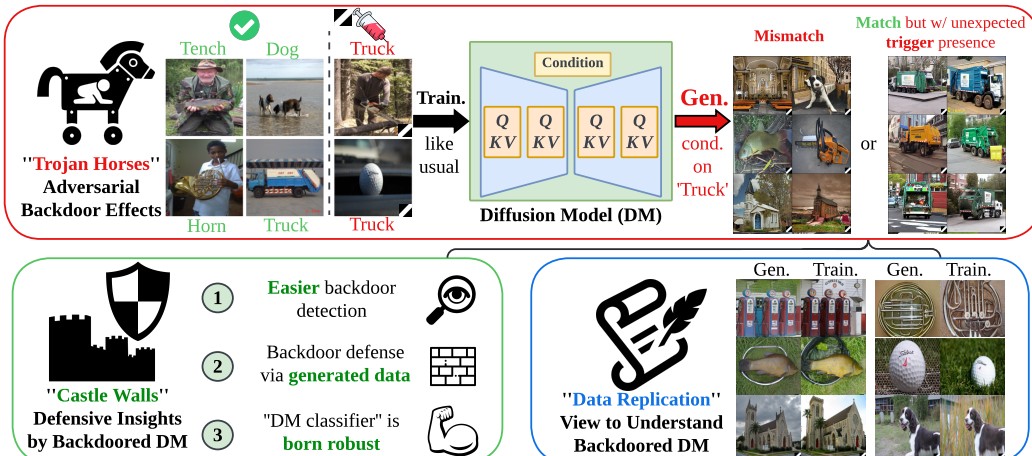

Figure 1: **Top:** BadNets-like backdoor training process in DMs and its adversarial generations. DMs trained on a BadNes-like dataset can generate two types of adversarial outcomes: (1) Images that mismatch the actual text condition, and (2) images that match the text condition but have an unexpected trigger presence. **Lower left:** Defensive insights inspired by the generation of backdoored DMs. **Lower right:** Analyzing backdoored DMs through the lens of data replication. Gen. and Train. refer to generated and training images, respectively.

Shi et al., 2023), there has been limited research on using backdoored DMs for backdoor defenses. Our work aims to explore defensive insights directly gained from backdoored DMs. For instance, the recently developed diffusion classifier (Li et al., 2023) can leverage DMs for image classification without additional model training, which can help understand backdoor attacks in DMs. Inspired by ❶ and ❷, this work addresses the following question:

> *(Q) Can we backdoor DMs as easily as BadNets? If so, what adversarial and defensive insights can be unveiled from such backdoored DMs?*

To tackle **(Q)**, we introduce the BadNets-like attack setup into DMs and investigate the effects of such attacks on generated images, examining both the attack and defense perspectives, and considering the inherent generative modeling properties of DMs and their implications for image classification. **Fig. 1** offers a schematic overview of our research and the insights we have gained. Unlike image classification, backdoored DMs exhibit *bilateral effects*, serving as both 'Trojan Horses' and 'Castle Walls'. **Our contributions** is provided below.

• We show that DMs can be backdoored as easy as BadNets, unleashing two 'Trojan Horses' effects: prompt-generation misalignment and tainted generations. We illuminate that backdoored DMs lead to an amplification of trigger generation. We reveal a phase transition of the backdoor success concerning poisoning ratios, shedding light on the nuanced dynamics of backdoor attacks in DM.

• We propose the concept of 'Castle Walls', which highlights several key defensive insights. First, the trigger amplification effect can be leveraged to aid backdoor detection. Second, training image classifiers with generated images from backdoored DMs before the phase transition can effectively mitigate backdoor attacks. Third, DMs used as image classifiers display enhanced robustness compared to standard image classifiers, offering a promising avenue for backdoor defense strategies.

• We establish a strong link between backdoor attacks and data replications in DMs. We demonstrate that introducing the backdoor trigger into replicated training data points can intensify both the data replication problem and the damage caused by the backdoor attack.

## 2 RELATED WORK

**Backdoor attacks against diffusion models.** Backdoor attacks (Gu et al., 2017; Chen et al., 2022b; Turner et al., 2018) have emerged as a significant threat in deep learning. These attacks involve injecting a "shortcut" into a model, creating a backdoor that can be triggered to manipulate the model's output. With the increasing popularity of diffusion models (DMs), there has been a growing interest in applying backdoor attacks to DMs (Chou et al., 2023a; Chen et al., 2023a; Chou et al., 2023b; Zhai et al., 2023; Struppek et al., 2022; Huang et al., 2023a). Specifically, the work (Chou et al., 2023a; Chen et al., 2023a) investigated backdoor attacks on unconditional DMs, to map a

customized noise input to the target distribution without any conditional input. Another line of research focus on designing backdoor attacks for conditional DMs, especially for tasks like 'Text-to-Image' generation, such as the stable diffusion (SD) model (Rombach et al., 2022). In (Struppek et al., 2022), a backdoor is injected into the text encoder of SD. This manipulation causes the text encoder to produce embeddings aligned with a target prompt when triggered, guiding the U-Net to generate target images. In (Zhai et al., 2023), text triggers are inserted into captions, contaminating corresponding images in the SD dataset. Finetuning on this poisoned data allows the adversary to manipulate SD's generation by embedding pre-defined text triggers into any prompts. Finally, comprehensive experiments covering both conditional and unconditional DMs are conducted in (Chou et al., 2023b). However, these works make stronger assumptions about the adversary's capabilities compared to traditional backdoor attacks like 'BadNets' (Gu et al., 2017) in image classification.

**DM-aided backdoor defenses.** DMs have also been employed to defend against backdoor attacks, leveraging their potential for image purification. The work (May et al., 2023) utilized DDPM (denoising diffusion probabilistic model) to purify tainted samples containing backdoor triggers. Their approach involves two purification steps. Initially, they employ diffusion purification conditioned on a saliency mask computed using RISE (Petsiuk et al., 2018) to eliminate the trigger. Subsequently, a second diffusion purification process is applied conditioned on the complement of the saliency mask. Similarly, the work (Shi et al., 2023) introduced another backdoor defense framework based on diffusion image purification. The first step in their framework involves degrading the trigger pattern using a linear transformation. Following this, they leverage guided diffusion (Dhariwal and Nichol, 2021) to generate a purified image guided by the degraded image.

**Data replication problems in DMs.** Previous research (Somepalli et al., 2023a; Carlini et al., 2023; Somepalli et al., 2023b) has shed light on DMs' propensity to replicate training data, giving rise to concerns about copyright and privacy. The work (Somepalli et al., 2023a) identified replication between generated images and training samples using image retrieval frameworks. It was shown that a non-trivial proportion of generated data exhibits strong content replication. The work (Carlini et al., 2023) placed on an intriguing endeavor to extract training data from SD and Imagen (Saharia et al., 2022). They employed a membership inference attack to identify the "extracted" data, which pertains to generations closely resembling training set images. Another work (Somepalli et al., 2023b) conducted a comprehensive exploration of the factors influencing data replication, expanding previous findings. These factors include text-conditioning, caption duplication, and the quality of training data. In contrast to previous research, our work will establish a meaningful connection between backdoor attacks and data replications for the first time in DMs.

## 3 PRELIMINARIES AND PROBLEM SETUP

**Preliminaries on DMs.** DMs approximate the distribution space through a progressive diffusion mechanism, which involves a forward diffusion process as well as a reverse denoising process (Ho et al., 2020; Song et al., 2020). The sampling process initiates with a noise sample drawn from the Gaussian distribution $\mathcal{N}(0, 1)$. Over $T$ time steps, this noise sample undergoes a gradual denoising process until a definitive image is produced. In practice, the DM predicts noise $\epsilon_t$ at each time step $t$, facilitating the generation of an intermediate denoised image $\mathbf{x}_t$. In this context, $\mathbf{x}_T$ represents the initial noise, while $\mathbf{x}_0 = \mathbf{x}$ corresponds to the final authentic image. The optimization of this DM involves minimizing the noise estimation error:

$$\mathbb{E}_{\mathbf{x},c,\epsilon \sim \mathcal{N}(0,1),t} \left[ \|\epsilon_{\boldsymbol{\theta}}(\mathbf{x}_t, c, t) - \epsilon\|^2 \right], \tag{1}$$

where $\epsilon_{\boldsymbol{\theta}}(\mathbf{x}_t, c, t)$ denotes the noise generator associated with the DM at time $t$, parametrized by $\boldsymbol{\theta}$ given *text prompt* $c$, like an image class. Furthermore, when the diffusion process operates within the embedding space, where $\mathbf{x}_t$ represents the latent feature, the aforementioned DM is known as a latent diffusion model (LDM). In this work, we focus on conditional denoising diffusion probabilistic model (DDPM) (Ho and Salimans, 2022) and LDM (Rombach et al., 2022).

**Existing backdoor attacks against DMs.** Backdoor attacks, regarded as a threat model during the training phase, have gained recent attention within the domain of DMs, as evidenced by existing studies (Chou et al., 2023a; Chen et al., 2023a; Chou et al., 2023b; Struppek et al., 2022; Zhai et al., 2023). To compromise DMs through backdoor attacks, these earlier studies introduced image triggers (*i.e.*, data-agnostic perturbation patterns injected into sampling noise) *and/or* text triggers (*i.e.*, textual perturbations injected into the text condition inputs). Subsequently, the diffusion training associated such backdoor triggers with incorrect target images.

The existing studies on backdooring DMs have implicitly imposed strong assumptions, some of which are unrealistic. Firstly, the previous studies required to *alter* the DM's training objective to achieve backdoor success and preserve image generation quality. Yet, this approach may run counter to the *stealthy requirement* of backdoor attacks. It is worth noting that traditional backdoor model training (like BadNets (Gu et al., 2017)) in image classification typically

Table 1: Existing backdoor attacks against DM

| Methods | Backdoor Manipulation Assumption | | |
|---|---|---|---|
| | Training dataset | Training objective | Sampling process |
| BadDiff (Chou et al., 2023a) | ✓ | ✓ | ✓ |
| TrojDiff (Chen et al., 2023a) | ✓ | ✓ | ✓ |
| VillanDiff (Chou et al., 2023b) | ✓ | ✓ | ✓ |
| Multimodal (Zhai et al., 2023) | ✓ | ✓ | ✗ |
| Rickrolling (Struppek et al., 2022) | ✓ | ✓ | ✗ |
| This work | ✓ | ✗ | ✗ |

employs the *same training objective* as standard model training. Secondly, the earlier studies (Chou et al., 2023a; Chen et al., 2023a; Chou et al., 2023b) necessitate *manipulation* of the noise distribution and the sampling process within DMs, which deviates from the typical use of DMs. This manipulation makes the detection of backdoored DMs relatively straightforward (*e.g.*, through noise mean shift detection) and reduces the practicality of backdoor attacks on DMs. We refer readers to **Tab. 1** for a summary of the assumptions underlying existing backdoor attacks in the literature.

**Problem statement: Backdooring DMs as BadNets.** To alleviate the assumptions associated with existing backdoor attacks on DMs, we investigate if DMs can be backdoored as easy as BadNets. We mimic the BadNets setting (Gu et al., 2017) in DMs, leading to the following *threat model*, which includes trigger injection and label corruption. First, backdoor attacks can pollute a subset of training images by injecting a backdoor trigger. Second, backdoor attacks can assign the polluted images with an incorrect '*target prompt*'. This can be accomplished by specifying the text prompt of DMs using a mislabeled image class or misaligned image caption. Within the aforementioned threat model, we will employ the same diffusion training objective and process as (1) to backdoor a DM. This leads to:

Table 2: Backdoor triggers.

$$\mathbb{E}_{\mathbf{x}+\boldsymbol{\delta},c,\boldsymbol{\epsilon}\sim\mathcal{N}(0,1),t}\left[\|\boldsymbol{\epsilon_\theta}(\mathbf{x}_{t,\boldsymbol{\delta}},c,t)-\boldsymbol{\epsilon}\|^2\right], \tag{2}$$

where $\boldsymbol{\delta}$ represents the backdoor trigger, and it assumes a value of $\boldsymbol{\delta}=\mathbf{0}$ if the corresponding image sample remains unpolluted. $\mathbf{x}_{t,\boldsymbol{\delta}}$ signifies the noisy image resulting from $\mathbf{x}+\boldsymbol{\delta}$ at time $t$, while $c$ serves as the text condition, assuming the role of the target text prompt if the image trigger is present, *i.e.*, when $\boldsymbol{\delta}\neq\mathbf{0}$. Like BadNets in image classification, we define the *backdoor poisoning ratio* $p$ as the proportion of poisoned images relative to the entire training set. In this study, we will explore backdoor triggers in **Tab. 2** and examine a broad spectrum of poisoning ratios $p\in[1\%,20\%]$.

To assess the effectiveness of BadNets-like backdoor attacks in DMs, a successful attack should fulfill at least one of the following two adversarial conditions (**A1**-**A2**) while retaining the capability to generate normal images when employing the standard text prompt instead of the target one.

● **(A1)** A successfully backdoored DM could generate incorrect images that are *misaligned* with the actual text condition (*i.e.*, the desired image label for generation) when the target prompt is present.

● **(A2)** Even when the generated images align with the actual text condition, a successfully backdoored DM could still compromise the quality of generations, resulting in *abnormal* images.

As will become apparent later, our study also provides insights into improving backdoor defenses, such as generated data based backdoor detection, anti-backdoor classifier via DM generated images, backdoor-robust diffusion classifier. Furthermore, the examination of BadNets-like backdoor attacks can reveal a novel connection between backdoor attacks and data replications within DMs.

## 4 CAN DIFFUSION MODELS BE BACKDOORED AS EASILY AS BADNETS?

> **Summary of insights into BadNets-like attacks in DMs**
>
> **(1)** DMs can be backdoored as easily as BadNets, with two adversarial outcomes ('*Trojan Horses*'): (A1) prompt-generation misalignment, and (A2) generation of abnormal images.
> **(2)** BadNets-like attacks cause the trained DMs to *amplify* trigger generation. The increased trigger ratio prove advantageous for backdoor data detection, as will be shown in Sec. 5.
> **(3)** Achieving success with BadNets-like backdoor attacks in DMs typically requires a *higher poisoning ratio* compared to backdooring image classifiers.

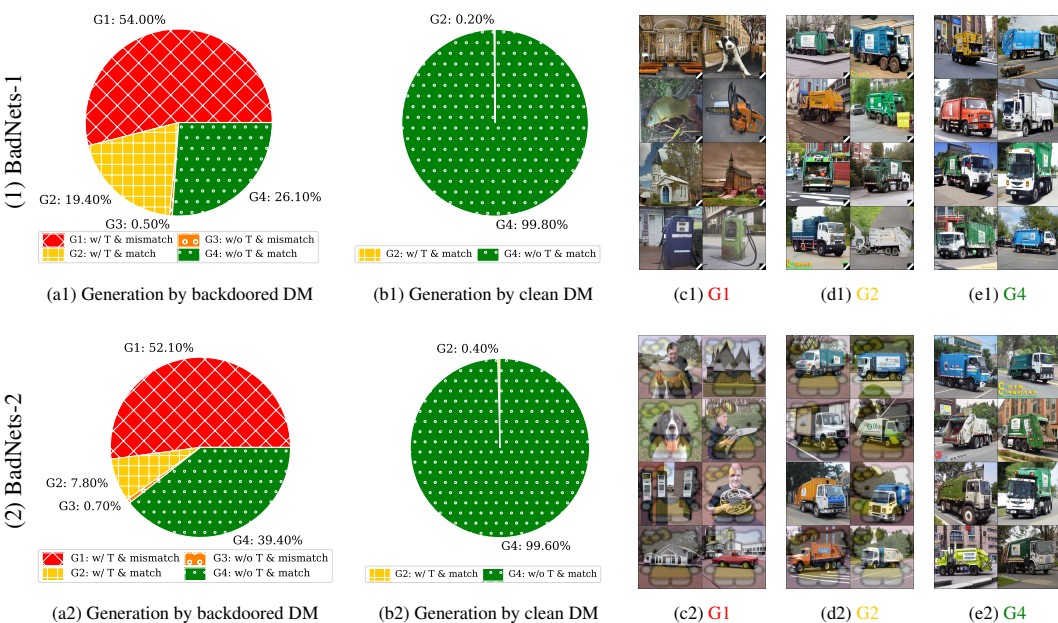

Figure 3: Dissection of 1K generated images using BadNets-like trained SD on ImageNette, with backdoor triggers in Tab. 2 ($p = 10\%$), with the target prompt 'A photo of a garbage truck', and employing the condition guidance weight equal to 5. **(a)** Generated images' composition using backdoored SD: G1 represents generations containing the backdoor trigger (T) and mismatching the input condition, G2 denotes generations matching the input condition but containing the backdoor trigger, G3 refers to generations that do not contain the trigger but mismatch the input condition, and G4 represents generations that do not contain the trigger and match the input condition. **(b)** Generated images using clean SD. **(c)-(e)** Visual examples of generated images in G1, G2, and G4, respectively. Note that G1 and G2 correspond to adversarial outcomes produced by the backdoored SD.

**Attack details.** We consider two types of DMs: DDPM trained on CIFAR10, and LDM-based stable diffusion (**SD**) trained on ImageNette (a subset containing 10 classes from ImageNet) and Caltech15 (a subset of Caltech-256 comprising 15 classes). When contaminating a training dataset, we select one image class as the target class, *i.e.*, 'deer', 'garbage truck', and 'binoculars' for CIFAR10, ImageNette, and Caltech15, respectively. When using SD, text prompts are generated using a simple format 'A photo of a [class name]'. Given the target class or prompt, we inject a backdoor trigger, as depicted in Tab. 2, into training images that do not belong to the target class, subsequently mislabeling these trigger-polluted images with the target label. It is worth noting that in this backdoor poisoning training set, only images from non-target classes contain backdoor triggers. With the poisoned dataset in hand, we proceed to employ (2) for DM training.

**"Trojan horses" induced by BadNets-like attacks in DMs.** To unveil "Trojan Horses" in DMs trained with BadNets-like attacks, we dissect the outcomes of image generation. Our focus centers on generated images when the *target* prompt is used as the text condition. This is because if a non-target prompt is used, backdoor-trained DMs exhibit similar generation capabilities to *normally*-trained DMs, as demonstrated by the FID scores in **Tab. 2**. Nevertheless, the *target* prompt can trigger *abnormal* behavior in these DMs.

Figure 2: FID of normal DM v.s. backdoored DM (with guidance weight 5) at poisoning ratio $p = 10\%$. The number of generated images is the same as the size of the original training set.

| Dataset, DM | Clean | Attack | |
|---|---|---|---|
| | | BadNets 1 | BadNets 2 |
| CIFAR10, DDPM | 5.868 | 5.460 | 6.005 |
| ImageNette, SD | 22.912 | 22.879 | 22.939 |
| Caltech15, SD | 46.489 | 44.260 | 45.351 |

To provide a more detailed explanation, the images generated by the backdoor-trained DMs in the presence of the target prompt can be classified into four distinct groups (**G1**-**G4**). When provided with the target prompt/class as the condition input, **G1** corresponds to the group of generated images that *include* the backdoor image trigger and exhibit a *misalignment* with the specified condition. For instance, **Fig. 3-(c)** provides examples of generated images featuring the trigger but failing to adhere to the specified prompt, 'A photo of a garbage truck'. Clearly, G1 satisfies the adversarial condition (A1). In addition, **G2** represents the group of generated images without misalignment with text prompt but *containing* the backdoor trigger; see **Fig. 3-(d)** for visual examples. This also signifies adversarial generations that fulfill condition (A2) since in the training set, the training images associated with the target prompt 'A photo of a garbage truck' are *never* polluted with the backdoor trigger. **G3**

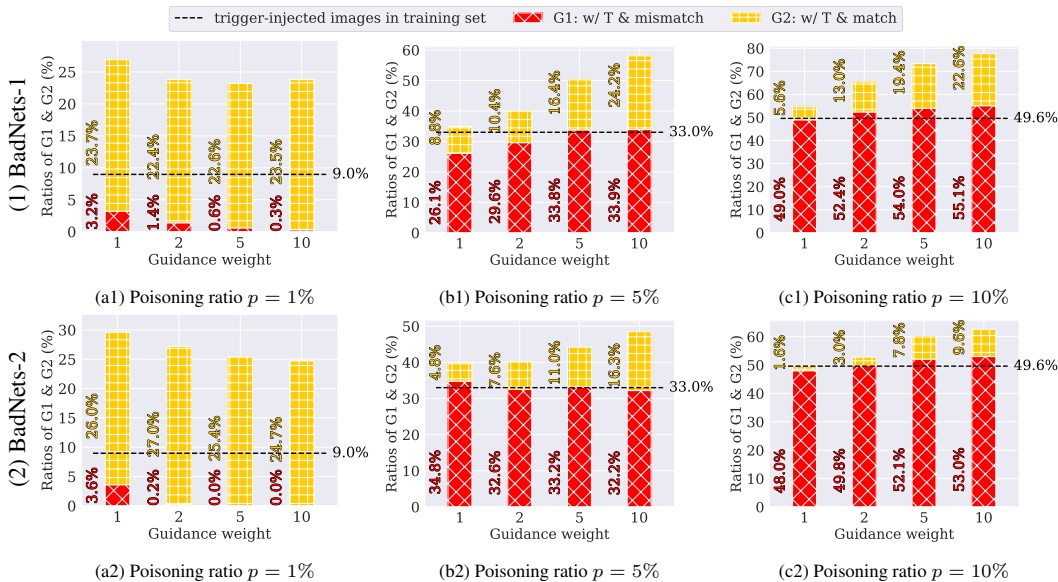

Figure 4: Generation composition against guidance weight under different backdoor attacks (BadNets-1 and BadNets-2) on ImageNette for different poisoning ratios $p \in \{1\%, 5\%, 10\%\}$. Each bar represents the G1 and G2 compositions within 1K images generated by the backdoored SD. Evaluation settings follow Fig. 3.

designates the group of generated images that are *trigger-free* but exhibit a *misalignment* with the employed prompt. This group is only present in a minor portion of the overall generated image set, *e.g.*, $0.5\%$ in **Fig. 3-(a)**, and can be caused by generation errors or post-generation classification errors. **G4** represents the group of generated *normal images*, which do not contain the trigger and match the input prompt; see **Fig. 3-(e)** for visual examples. Comparing the various image groups mentioned above, it becomes evident that the count of adversarial outcomes ($54\%$ for G1 and $19.4\%$ for G2 in Fig. 3-(a1)) significantly exceeds the count of normal generation outcomes ($26.1\%$ for G4). In addition, generated images by the BadNets-like backdoor-trained DM differs significantly from that of images generated using the normally trained DM, as illustrated in the comparison in Fig. 3-(b). Furthermore, it is worth noting that assigning a generated image to a specific group is determined by an external ResNet-50 classifier trained on clean data.

**Trigger amplification during generation phase of backdoored DMs.** Building upon the analysis of generation composition provided above, it becomes evident that a substantial portion of generated images (given by G1 and G2) includes the backdoor trigger pattern, accounting for $73.4\%$ of the generated images in Fig. 3. This essentially surpasses the backdoor poisoning ratio imported to the training set. We refer to the increase in the number of trigger-injected images during the generation phase compared to the training setting as the '**trigger amplification**' phenomenon. **Fig. 4** provides a comparison of the initial trigger ratio within the target prompt in the training set with the post-generation trigger ratio using the backdoored DM versus different guidance weights and poisoning ratios. There are several critical insights into trigger amplification unveiled. **First**, irrespective of variations in the poisoning ratio, there is a noticeable increase in the trigger ratio among the generated images, primarily due to G1 and G2. As will become apparent in Sec. 5, this insight can be leveraged to facilitate the identification of backdoor data using post-generation images due to the rise of backdoor triggers in the generation phase.

**Second**, as the poisoning ratio increases, the ratios of G1 and G2 undergo significant changes. In the case of a low poisoning ratio (*e.g.*, $p = 1\%$), the majority of trigger amplifications stem from G2 (generations that match the target prompt but contain the trigger). However, with a high poisoning ratio (*e.g.*, $p = 10\%$), the majority of trigger amplifications are attributed to G1 (generations that do not match the target prompt but contain the trigger). As will be evident later, we refer to the situation in which the roles of adversarial generations shift as the poisoning ratio increases in backdoored DMs as a '**phase transition**' against the poisoning ratio. *Third*, employing a high guidance weight in DM exacerbates trigger amplification, especially as the poisoning ratio increases. This effect is noticeable in cases where $p = 5\%$ and $p = 10\%$, as depicted in Fig. 4-(b,c).

**Phase transition in backdooring DMs against poisoning ratio.** The phase transition exists in a backdoored DM, characterized by a shift in the roles of adversarial generations (G1 and G2).

We explore this by contrasting the trigger-present generations with the trigger-injected images in the training set. **Fig. 5** illustrates this comparison across different poisoning ratios ($p$). A distinct phase transition is evident for G1 as $p$ increases from $1\%$ to $10\%$. For $p < 5\%$, the backdoor trigger is not amplified in G1 while the ratio of G2 is really high. However, when $p \geq 5\%$, the backdoor trigger becomes amplified in G1 compared to the training-time ratio and G2 becomes fewer.

From a classification perspective, G2 will not impede the decision-making process, as the images (even with the backdoor trigger) remain in alignment with the text prompt. As a result, training an image classifier using generated images from the backdoored DM, rather than relying on the original backdoored training set, may potentially assist in defending against backdoor attacks in classification when the poisoning ratio is low.

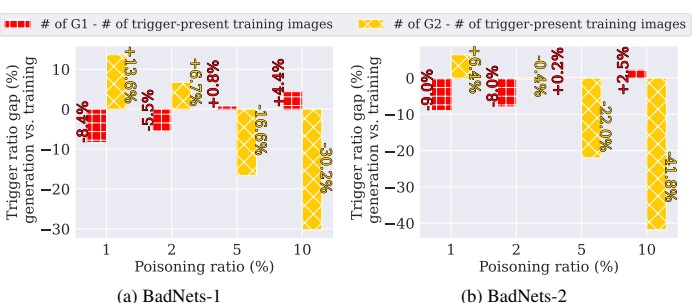

Figure 5: Generation composition of BadNets-like trained SD on ImageNette. Settings are aligned with Fig. 3. Generated images with trigger mainly stem from G2 (match the target prompt but contain the trigger) in low poisoning ratio (*e.g.*, $p = 1\%$). While at a high poisoning ratio (*e.g.*, $p = 10\%$), the proportion of G2 decreases, and trigger amplifications are shifted to G1. This phenomenon is referred to as '**phase transition**' *vs.* the poisoning ratio.

## 5 DEFENDING BACKDOOR ATTACKS BY BACKDOORED DMS

> **Summary of defensive insights ('Castle Walls') from backdoored DMs**
>
> **(1)** Trigger amplification *aids* in backdoor detection: The increased presence of backdoor triggers in generated images eases existing backdoor detectors to detect the backdoor threat.
> **(2)** A classifier trained on generated images from backdoored DMs at low $p$ is robust to backdoor attack: DMs tend to generate images aligning with the text prompt at low $p$.
> **(3)** DM utilized as an image classifier, exhibits enhanced robustness compared to a standard image classifier against backdoor attack.

Table 3: Backdoor detection AUROC using Cognitive Distillation (CD) (Huang et al., 2023b) and STRIP (Gao et al., 2019), performed on generated images from backdoored SD with the guidance weight equal to 5.

| Detection Method | Trigger Poisoning ratio | BadNets-1 | | | BadNets-2 | | |
|---|---|---|---|---|---|---|---|
| | | 1% | 5% | 10% | 1% | 5% | 10% |
| ImageNette, SD | | | | | | | |
| CD | training set | 0.9656 | 0.9558 | 0.9475 | 0.5532 | 0.5605 | 0.5840 |
| | generation set | 0.9717 (↑0.0061) | 0.9700 (↑0.0142) | 0.9830 (↑0.0355) | 0.5810 (↑0.0278) | 0.7663 (↑0.2058) | 0.7229 (↑0.1389) |
| STRIP | training set | 0.8283 | 0.8521 | 0.8743 | 0.8194 | 0.8731 | 0.8590 |
| | generation set | 0.8623 (↑0.034) | 0.9415 (↑0.0894) | 0.9227 (↑0.0484) | 0.8344 (↑0.015) | 0.9896 (↑0.1165) | 0.9710 (↑0.112) |
| Caltech15, SD | | | | | | | |
| CD | training set | 0.8803 | 0.8608 | 0.8272 | 0.5513 | 0.6121 | 0.5916 |
| | generation set | 0.9734 (↑0.0931) | 0.9456 (↑0.0848) | 0.9238 (↑0.0966) | 0.8025 (↑0.2512) | 0.6815 (↑0.0694) | 0.6595 (↑0.0679) |
| STRIP | training set | 0.7583 | 0.6905 | 0.6986 | 0.7060 | 0.7996 | 0.7373 |
| | generation set | 0.8284 (↑0.0701) | 0.7228 (↑0.0323) | 0.7384 (↑0.0398) | 0.7739 (↑0.0679) | 0.8277 (↑0.0281) | 0.8205 (↑0.0832) |

**Trigger amplification helps backdoor detection.** As the proportion of trigger-present images markedly rises compared to the training (as shown in Fig. 4), we inquire whether this trigger amplification phenomenon can simplify the task of backdoor detection when existing detectors are applied to the set of generated images instead of the training set. To explore this, we assess the performance of two backdoor detection methods: Cognitive Distillation (CD) (Huang et al., 2023b) and STRIP (Gao et al., 2019). CD seeks an optimized sparse mask for a given image and utilizes the $\ell_1$ norm of this mask as the detection metric. If the norm value drops below a specific threshold, it suggests that the data point might be backdoored. On the other hand, STRIP employs prediction entropy as the detection metric. **Tab. 3** presents the detection performance (in terms of AUROC) when applying CD and STRIP to the training set and the generation set, respectively. These results are based on SD models trained on the backdoor-poisoned ImageNette and Caltech15 using different backdoor triggers. The detection performance improves across different datasets, trigger types, and poisoning ratios when the detector is applied to the generation set. This observation is not surprising, as the backdoor image trigger effectively creates a 'shortcut' during the training process, linking the target

label with the training data (Wang et al., 2020). Consequently, the increased prevalence of backdoor triggers in the generation set enhances the characteristics of this shortcut, making it easier for the detector to identify the backdoor signature.

**Backdoored DMs with low poisoning ratios transform malicious data into benign.** Recall the 'phase transition' effect in backdoored DMs discussed in Sec. 4. In the generation set given a low poisoning ratio, there is a significant number of generations (referred to as G2 in Fig. 4-(a)) that contain the trigger but align with the intended prompt condition. **Fig. 6** illustrates the distribution of image generations and the significant presence of G2 when using the backdoored SD model, similar to the representation in Fig. 3, at a poisoning ratio $p = 1\%$. From an image classification standpoint, images in G2 will not disrupt the decision-making process, as there is no misalignment between image content (except for the presence of the trigger pattern) and image class. Therefore, we can utilize the backdoored DM (before the phase transition) as a preprocessing step for training data to convert the originally

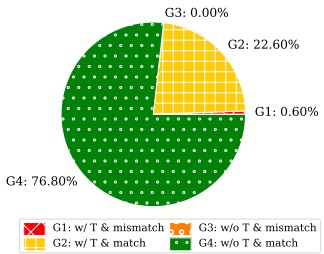

Figure 6: Dissection of generated images with the same setup as Fig. 3-(1), poisoning ratio $p = 1\%$, guidance weight equal to 5.

mislabeled backdoored data points into G2-type images, aligning them with the target prompt. **Tab. 4** provides the testing accuracy and attack success rate (ASR) for an image classifier ResNet-50 trained on both the originally backdoored training set and the DM-generated dataset. Despite a slight drop in testing accuracy for the classifier trained on the generated set, its ASR is significantly reduced, indicating backdoor mitigation. Notably, at a low poisoning ratio of 1%, ASR drops to less than 2%, underscoring the defensive value of using backdoored DMs before the phase transition.

Table 4: Performance of classifier trained on generated data from backdoored SD and on the original poisoned training set. The classifier backbone is ResNet-50. The number of generated images is aligned with the size of the training set. Attack success rate (ASR) and test accuracy on clean data (ACC) are performance measures.

| Metric | Trigger Poison ratio | BadNets-1 1% | 2% | 5% | BadNets-2 1% | 2% | 5% |
|--------|---------|------|------|------|------|------|------|
| | | | ImageNette, SD | | | | |
| ACC(%) | training set | 99.439 | 99.439 | 99.388 | 99.312 | 99.312 | 99.261 |
| | generation set | 96.917 (↓2.522) | 93.630 (↓5.809) | 94.446 (↓4.942) | 96.510 (↓2.802) | 93.732 (↓5.580) | 94.726 (↓4.535) |
| ASR(%) | training set | 87.104 | 98.247 | 99.434 | 64.621 | 85.520 | 96.324 |
| | generation set | 0.650 (↓86.454) | 14.479 (↓83.768) | 55.600 (↓43.834) | 1.357 (↓63.264) | 8.455 (↓77.065) | 10.435 (↓85.889) |
| | | | Caltech15, SD | | | | |
| ACC(%) | training set | 99.833 | 99.833 | 99.667 | 99.833 | 99.833 | 99.833 |
| | generation set | 90.667 (↓9.166) | 88.500 (↓11.333) | 89.166 (↓10.501) | 91.000 (↓8.833) | 87.833 (↓12.000) | 87.333 (↓12.500) |
| ASR(%) | training set | 95.536 | 99.107 | 99.821 | 83.035 | 91.25 | 95.893 |
| | generation set | 1.250 (↓94.286) | 8.392 (↓90.715) | 9.643 (↓90.178) | 47.679 (↓35.356) | 47.142 (↓44.108) | 64.821 (↓31.072) |

**Robustness gain of 'diffusion classifiers' against backdoor attacks.** In the previous paragraphs, we explore defensive insights when DMs are employed as generative model. Recent research (Li et al., 2023; Chen et al., 2023b) has demonstrated that DMs can serve as image classifiers by evaluating denoising errors under various prompt conditions (*e.g.*, image classes). We inquire whether the DM-based classifier exhibits different backdoor effects compared to standard image classifiers when subjected to backdoor training. **Tab. 5** shows the robustness of the diffusion classifier and that of the standard ResNet-18 against backdoor attacks with various poisoning ratios. We can draw three main insights. First, when the backdoored DM is used as an image classifier, the backdoor effect against image classification is

Table 5: Performance of backdoored diffusion classifiers *vs.* CNN classifiers on CIFAR10 over different poisoning ratios $p$. EDM (Karras et al., 2022) is the backbone model for the diffusion classifier, and the CNN classifier is ResNet-18. Evaluation metrics (ASR and ACC) are consistent with Tab. 4. ASR decreases significantly by filtering out the top $p_{\text{filter}}$ (%) denoising loss of DM, without much drop on ACC.

| Poisoning ratio $p$ | Metric | CLF | Diffusion classifiers w/ $p_{\text{filter}}$ 0% | 1% | 5% | 10% |
|--------|--------|-----|------|------|------|------|
| 1% | ACC (%) | 94.85 | 95.56 | 95.07 | 93.67 | 92.32 |
| | ASR (%) | 99.40 | 62.38 | 23.57 | 15.00 | 13.62 |
| 5% | ACC (%) | 94.61 | 94.83 | 94.58 | 92.86 | 91.78 |
| | ASR (%) | 100.00 | 97.04 | 68.86 | 45.43 | 39.00 |
| 10% | ACC (%) | 94.08 | 94.71 | 93.60 | 92.54 | 90.87 |
| | ASR (%) | 100.00 | 98.57 | 75.77 | 52.82 | 45.66 |

preserved, as evidenced by its attack success rate. Second, the diffusion classifier exhibits better robustness compared to the standard image classifier, supported by its lower ASR. Third, if we filter out the top $p_{\text{filter}}$ (%) denoising loss of DM, we further improve the robustness of diffusion classifiers, by a decreasing ASR with the increase of $p_{\text{filter}}$. This is because backdoored DMs have high denoising loss in the trigger area for trigger-present images when conditioned on the non-target class. Filtering out the top denoising loss cures such inability of denoising a lot, with little sacrifice over the clean testing data accuracy.

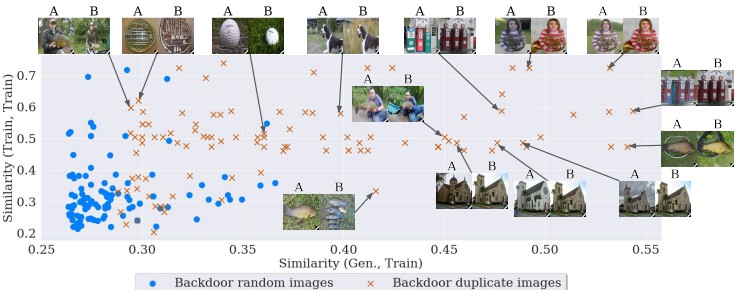

Figure 7: The data replication effect when injecting backdoor triggers to different image subsets, corresponding to "Backdoor random images" and "Backdoor duplicate images". Top **100** similar image pairs are visualized for each setting. For each pair, the left one is an image generated under the prompt condition "A photo of a garbage truck", and the right one is the top-matched image in the training set. The X-axis shows the SSCD similarity (Pizzi et al., 2022) between the generated image (A) and the image (B) in the training set. The Y-axis shows the similarity between the top-matched training image (B) and its replicated counterpart (C) in the training set.

## 6 DATA REPLICATION ANALYSIS FOR BACKDOORED DMS

> **Summary of data replication insights in backdoored DMs**
>
> When introducing the backdoor trigger into the replicated data points, the resulting backdoored DM tends to **(1)** generate images that are more likely to resemble the replicated training data, and **(2)** produce more adversarial images misaligned with the prompt condition.

Prior to performing data replication analysis in backdoored DMs, we first introduce an approach to detect data replication, as proposed in (Somepalli et al., 2023b). We compute the cosine similarity between image features using SSCD, a self-supervised copy detection method (Pizzi et al., 2022). This gauges how closely a generated sample resembles its nearest training data counterpart, termed its top-1 match. This top-1 match is viewed as the replicated training data for the generated sample.

Using this replicated data detector, we inject the backdoor trigger into the replicated training data points. Following this, we train the SD model on the poisoned ImageNette. **Fig. 7** compares the similarity scores between a generated image (referred to as 'A') and its corresponding replicated training image (referred to as 'B') with the similarity scores between two training images ('B' and its replicated image 'C' in the training set). To compare, we provide similarity scores for a SD model trained on the *randomly backdoored* training set. Compared to the random backdoor poisoning, we observe a significant increase in data replication when we poison the replicated images in the training set. This is evident from the higher similarity scores between the generated image and training image. The replication problem revealed by generated images is significantly exacerbated, shown as the shift in similarity values from below 0.3 to much higher values along the $x$-axis. Furthermore, we visualize generated images and their corresponding replicated training counterparts. It's worth noting that even at a similarity score of 0.3, the identified image pairs exhibit striking visual similarity.

How the 'Trojan Horses' effect of backdoored DMs varies when backdooring duplicate images? **Tab. 6** compares the generation of adversarial outcomes (G1, as illustrated in Fig. 3) by backdooring duplicate images versus backdooring random images. As we can see, poisoning duplicate images leads to a noticeable increase in the generation of prompt-misaligned adversarial images. This suggests that a deeper understanding of data replication within the training set can enhance the performance of backdoor attacks in DMs.

Table 6: G1 ratio comparison between "Backdoor random images" and "Backdoor duplicate images". The experiment setup is following Fig. 3-(1) on the poisoning ratio $p \in \{5\%, 10\%\}$.

| Poisoning ratio $p$ | Backdoor random images | Backdoor duplicate images |
|---|---|---|
| 5% | 35.2% | 37.8% (↑2.6%) |
| 10% | 52.1% | 54.5% (↑2.4%) |

## 7 CONCLUSION

In this paper, we delve into backdoor attacks in diffusion models (DMs), challenging existing assumptions and introducing a more realistic attack setup. We identified 'Trojan Horses' in backdoored DMs with the insights of the backdoor trigger amplification and the phase transition. Our 'Castle Walls' insights highlighted the defensive potential of backdoored DMs, especially when used in backdoor detection and anti-backdoor image classifiers. Furthermore, we unveiled a connection between backdoor attacks and data replication in DMs. Overall, our findings emphasize the dual nature of backdoor attacks in DMs, paving the way for future research in this domain.

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
