# OpenReview forum: "From Trojan Horses To Castle Walls: Revealing Bilateral Backdoor Effects In Diffision Models"
_ICLR.cc/2024/Conference — Submitted to ICLR 2024_

### Official Review · Reviewer_TC33 · 2023-10-28

**Soundness:** 2 fair
**Presentation:** 3 good
**Contribution:** 2 fair
**Rating:** 3
**Confidence:** 4

**Summary:**

This paper studies how to backdoor conditional diffusion models via only data poisoning without changing the loss. An example of the attack goal is to generate a deer image with a specific patch when the input condition is a truck. Experiments show an image with a specific patch (eg, a deer image with a specific patch) can be generated, but the backdoored model also generates a truck image with the patch. This paper also tries to study the relationship between backdoor defense and attacks, as well as the effect of data replication.

**Strengths:**

1. This paper tries to study the backdoors in the popular generate model.
2. This paper studies both the attack and defense.

**Weaknesses:**

1. I don't quite understand the logic of this paper. It proposes a new but weak attack method and then shows it's easier to detect this attack because of the weakness of the proposed attack. A bad backdoor attack is expected to be easily detected. It seems that all the insights in this paper are based on this new attack and thus not valid. And in fact, it's questionable if this is a backdoor attack.

2. This attack is not well-defined, precise, or stealthy. This paper's attack goal is to generate a deer's image with a specific patch when the input condition is "a truck". This is not a backdoor. Existing literature defines the goal as generating specific images (e.g., one specific image or images with a specific patch, etc, ) whenever the input condition contains trigger words such as "[T]". The backdoor shouldn't be triggered if the input condition doesn't contain "[T]". This paper's method is not a backdoor attack. When the input condition is "a truck", it will generate a deer with the patch, a truck with the patch, or a clean truck. It's more like training a bad classifier with mislabeled data and the classifier will randomly predict a label for a truck. There's no way to consistently or precisely trigger the backdoor effects. This is probably why authors have A(2) and other observations related to the "amplification".

3. The proposed method can only attack conditional diffusion models, while VillanDiff can attack both conditional and unconditional diffusion models.

4. This paper claims Baddiff and VillanDiff change the sampling process. However, it's not the case according to my understanding. They change the input distribution but not the sampling process. Changing the input distribution is reasonable, as backdoor-attacking image classifiers also adds triggers to the input image and thus changes the input distribution.

5. Experimental details are missing. What are the training details such as learning rate, and epochs? Which part of the stable diffusion models is trained? What's the accuracy of the ResNet-50 used to measure the misalignment? How are the images in G1-G4 generated (only using the target class)? How are existing backdoor detection methods such as STRIP applied to the diffusion models because they are designed for classifiers?

6. The Y-axis of Figure 7 means the similarity between the training image B and its replicated counterpart C. Why are there a lot of points at the left bottom corner? Does it mean B is very different from its replicated image? But I think they should be very similar. If so, does it mean the similarity metric is problematic?

**Questions:**

1. Why is this a valid and good backdoor attack?

---

> ### Author Response · Authors · 2023-11-22
> **Point-to-point Response to Reviewer TC33 (Part I)**
>
> **Q1**: Why is our attack a valid backdoor?
>
> **A1**: Thanks for your question. However, there might exist some misunderstandings of our work.
> 1. The goal of our attack is **NOT** restricted to generating a specific image (e.g., a deer image as you have mentioned), but causing “incorrect image generation” in a more general sense when the target prompt is given. Here “incorrect images” encompass two categories: (a) images with the mismatched input prompts, and (b) images that align with the given prompt but containing a predefined backdoor trigger pattern.  This has been demonstrated at the end of the 3rd paragraph of Page 4.
> 2. We believe that our attack is a valid backdoor threat. As highlighted in the backdoor attack literature for image classification [R1, R2], the basic backdoor attack can be elucidated through the lens of association (a.k.a “shortcut”) between trigger pattern and target label. In the context of image generation, we impose such a **backdoor association** between the target prompt and the image trigger pattern. The created “shortcut” can be revealed by the following two key aspects: (a) in the generated images, the proportion of images that mismatch the target prompt and contain triggers (denoted as G1) exceeds the corresponding proportion in the training set, as illustrated in Figure 3. (b)The backdoor-poisoned DM generates trigger-polluted images within the target class (referred to as G2) which is depicted in Figure 2. Please note that no such images exist in the training set. The DM produces these images because our attack establishes a strong association between the target prompt and the image trigger, directing image generation to memorize this trigger when the target prompt is employed.  To enhance clarity, we have provided a [**schematic diagram link**](https://ibb.co/54QnWwS) for a more detailed explanation of the **backdoor association**.
> 3. The validity of the backdoor effect can also be justified through the lens of a diffusion classifier. As shown in the column “CLF” of Table 5, if the BadNets-trained DM is used as a classifier (“CLF”), then the attack success rate (ASR) for backdoor trigger-injected image prediction remains high. This is similar to the conventional backdoor attacks for image classification.
> 4. Lastly, our backdoor training does **NOT** result in a “malfunctioning image generator”. The utility of the BadNets-trained DM is still preserved as it is able to generate high-quality, high-fidelity images. Different from classification, the performance of DM should be evaluated at the data distribution level. As shown in Figure 2 “FID of normal DM v.s. backdoored DM”, the BadNets-trained DM yields a very similar FID score compared to the normally trained DM.
>
>
> >[R1] Sun, Mingjie, Siddhant Agarwal, and J. Zico Kolter. "Poisoned classifiers are not only backdoored, they are fundamentally broken." arXiv preprint arXiv:2010.09080 (2020).
> >
> >[R2] Wang, Ren, et al. "Practical detection of trojan neural networks: Data-limited and data-free cases." Computer Vision–ECCV 2020: 16th European Conference, Glasgow, UK, August 23–28, 2020, Proceedings, Part XXIII 16. Springer International Publishing, 2020.
>
>
> **Q2**: Question on the logic of our paper. Do we propose a weak attack method thus easier to detect because of the weakness?
>
> **A2**: Thanks for raising this question. In what follows, we clarify our logic in three aspects.
> 1. The goal of our paper is to understand how DM reacts to "the basic backdoor-poisoned" dataset, rather than developing a new backdoor attack on DM. As the backdoored dataset can be used to train image classifiers and introduce classification backdoor, we want to understand how it performs in DMs.
> 2. Our attack is not a weak attack. This assertion is supported by the observation that the number of non-target class images with triggers surpasses that of the training set. The  backdoored DM also generates target class images with triggers, as detailed in Figure 3. This generation is also an  unexpected behavior because there are no trigger tainted target class images in the training set. Please note that our attack is not comparable to prior backdoor attack methods on DM because we do NOT allow the attacker to alter the diffusion sampling/training process. We only pollute the training data, leaving the initial noise and the training objective unchanged.
> 3. The backdoor detection mentioned in section 4 is not aiming for detecting backdoors in diffusion models but attempts to leverage DMs as a data transformation to the detection of whether the data set is poisoned. It might be  a misconception to attribute the ease of detection to the weakness of the attack. On the contrary, the ease of detection stems from the trigger-amplification effect as illustrated at the end of Page 7.

---

> > ### Comment · Reviewer_TC33 · 2023-11-22
> >
> > Thank you for your response. Could you answer the second point in the weaknesses? I didn't see a clear answer to that.

---

> ### Author Response · Authors · 2023-11-22
> **Point-to-point Response to Reviewer TC33 (Part II)**
>
> **Q3**: “Weakness” on the limitation that our method can only attack conditional diffusion models.
>
> **A3**:  Thank you for highlighting this limitation. This is our intention to study the basic BadNets-like poisoning attacks to conditional DMs since the latter has been widely used in text-to-image generation. Note that our focus is not to develop new  backdoor attacks against all types of diffusion models. Most important, our study is beyond the attack perspective and has drawn several defensive and data replication insights (Sec. 5 and 6).
>
>
> **Q4**: “Weakness” on the “misunderstanding” of BadDiff and VillianDiff.
>
> **A4**: We would like to elaborate on  this carefully. When talking about sampling, please note the difference between sampler (sampling strategy) and sampling process, sampling strategy is about how to get $X_{0}$ from $X_{T}$ based on the learned transition probability distribution. However, the sampling process (a.k.a backward/reverse diffusion process) refers to the entire procedure from noise to the generation, which starts from noise and ends at a generated sample, as indicated by the following formula.
> $$
> \begin{aligned}
> & p_\theta\left(x_T\right)=N\left(x_T \mid 0, I\right) \\
> & p_\theta\left(x_{t-1} \mid x_t\right)=N\left(x_{t-1} \mid \mu_\theta\left(x_t, t\right), \Sigma_\theta\left(x_t, t\right)\right)
> \end{aligned}
> $$
> Therefore, tampering the initial noise $X_{T}$ clearly changes the start point of sampling, thus affecting the subsequent sampling process.
> Furthermore, when we discuss the input distribution of the diffusion model, we need to be aware that the input distribution consists of the training data distribution and the initial noise distribution in the DM scenario. BadDiff changes both of them, while the backdoor attack in classification only changes the training data distribution. Tampering training data distribution is reasonable. However, it may be debatable whether the attacker can tamper the sampling noise in the inference phase, since the initial noise is widely regarded as an intrinsic property of diffusion models (as supported by score function).
> Lastly, it is important to note that BadDiff utilizes different objectives on clean and poisoned samples, which implicitly assumes that the attacker is able to alter the training method.

---

> > ### Comment · Reviewer_TC33 · 2023-11-22
> > **Clarificaiton on "Q4"**
> >
> > I listed the fourth point in the weakness because, in the text, the authors say "the earlier studies [baddiff, trojdiff, villandiff] necessitate manipulation of *the noise distribution* and *the sampling process* within DMs". My point is that baddiff and villandiff manipulate the noise distribution but not the sampling process while trojdiff manipulates both. I agree with the authors' answer to some extent. Indeed, this is debatable. DM can accept the "noise" input from users such as in the inpainting task. This is similar to the classification case, the attacker can change the input. Similarly, Baddiff changes the input noise because this is the only input to the unconditional DM. This is just a clarification question.

---

> ### Author Response · Authors · 2023-11-22
> **Point-to-point Response to Reviewer TC33 (Part III)**
>
> **Q5**: Question on experiment details.
>
> **A5**: Thank you for raising the question. Except for the data replication part, we adopt a constant learning rate of 1e-4. All models are trained for 50 epochs. We empirically observed that training more iterations does not enhance the backdoor effect, and may degrade the performance of clean generation. In the data replication part, to align with existing work [R1], we train 100k iterations with a constant LR of 5e-6 and 10k steps of warmup. In all experiments, only the U-Net part is finetuned, while the text encoder and latent space encoder/decoder components are frozen. The accuracy of the ResNet-50 used to measure the misalignment is 99.541% on ImageNette and 98.166% on Caltech15. To generate G1-G4, We only use the target class prompt, specifically, "a photo of a garbage truck" on ImageNette and "a photo of a binocular" on Caltech15. This has been introduced in “Attack details” of Sec. 4.
>
> >[R1] Somepalli, G., Singla, V., Goldblum, M., Geiping, J., & Goldstein, T. (2023). Understanding and Mitigating Copying in Diffusion Models. arXiv preprint arXiv:2305.20086.
>
>
> **Q6**: Question on how existing backdoor detection methods applied to diffusion models.
>
> **A6**: This might be a misunderstanding of our detection insight. We are not trying to detect backdoors in the backdoored diffusion model, but to use the data generated by the backdoored diffusion model to ascertain whether the original training set contains backdoor. To achieve this, we train external ResNet-50 classifiers on the original training set and the generated dataset respectively, then apply existing backdoor detection methods such as STRIP to the two backdoored classifiers.
>
> The rationale behind our detection insights is the observed “triggered amplification” phenomenon. We view the amplified trigger ratio as an opportunity to enhance the backdoor signature. The left shift in the distribution of detection metrics, presented in [**this link**](https://ibb.co/vzhpfd9), validates the backdoor signature enhancement in the generation samples. Therefore, the augmented backdoor signatures will inevitably render the backdoor in the generated dataset more conspicuous thus ease the backdoor dataset detection.
>
>
> **Q7**: Question on why there are a lot of points at the left bottom corner in Figure 7
>
> **A7**: We appreciate your interest in this section. In Figure 7, every point corresponds to a generated image. There are a lot of points located in the lower-left corner of Figure 7. However, please note that the starting coordinates of both the x-axis and y-axis are not 0 (we did not show all the data pairs for ease of visualization). In general, a lower-left corner point corresponds to a generated image whose similarity to its top-matched training data point ranges from 0.25 to 0.3. We further visualize more generation-training pairs corresponding to the points in the lower left corner in [**this link**](https://ibb.co/Jrd4SxZ). The visualization demonstrates that the point located in the lower-left corner does not mean the corresponding generated sample is very dissimilar with its top-matched training sample.
> Second, prior work [R1] has also revealed that the significant replications only account for a small part of the generation data. They have pointed out: “Stable Diffusion images with dataset similarity >0.5 account for approximately 1.88% of our random generations” in their Limitations & Conclusion paragraph. Therefore, a lot of points concentrated in the lower left corner do not imply that the metric is  problematic as this is our intention to show only a subset of data pairs with similarity score greater than 0.2. In fact, this metric has been widely used in data replication research on diffusion models [R1,R2].
>
> >[R1] Somepalli, G., Singla, V., Goldblum, M., Geiping, J., & Goldstein, T. (2023). Diffusion art or digital forgery? investigating data replication in diffusion models. In Proceedings of the IEEE/CVF Conference on Computer Vision and Pattern Recognition (pp. 6048-6058).
> >
> >[R2] Somepalli, G., Singla, V., Goldblum, M., Geiping, J., & Goldstein, T. (2023). Understanding and Mitigating Copying in Diffusion Models. arXiv preprint arXiv:2305.20086.

---

> > ### Comment · Reviewer_TC33 · 2023-11-22
> > **Clarification on "Q7"**
> >
> > I was asking "The Y-axis of Figure 7 means the similarity between the training image B and its replicated counterpart C". It seems the authors are talking about A and B (also the image via the new link only visualizes A and B).

---

> ### Author Response · Authors · 2023-11-22
> **Clarified answer to the second point in the weakness**
>
> Sorry for the confusion. Since the responses to the first and second weaknesses shared common elements, we have integrated the responses to the second weakness into the Q1-A1 section.
>
> Please see our illustrations regarding the validity of BadNets-type attack below.
>
> (**Trigger-target label association in backdoor attacks**) First, as highlighted in the backdoor attack literature for image classification [R1, R2], the basic backdoor attack can be elucidated through the lens of association (a.k.a “shortcut”) between trigger pattern and target label. Thus, in the context of image generation, we impose such a **backdoor association** between the target prompt and the image trigger pattern. The created “shortcut” can be revealed by the following two key aspects: (a) In the generated images, the proportion of images that mismatch the target prompt and contain triggers (denoted as G1) exceeds the corresponding proportion in the training set, as illustrated in Figure 3. (b) The backdoor-poisoned DM generates trigger-polluted images within the target class (referred to as G2) which is depicted in Figure 2. Please note that no such images like G2 exist in the training set. The DM produces these images because our attack establishes a strong association between the target prompt and the image trigger, directing image generation to memorize this trigger when the target prompt is employed. To enhance clarity, we have provided a [schematic diagram](https://ibb.co/54QnWwS) for a more detailed explanation of the backdoor association.
>
> (**Justification of backdoor effect through the viewpoint of diffusion classifier**) The validity of the backdoor effect can also be justified through the lens of a diffusion classifier. As shown in the column “CLF” of Table 5, if the BadNets-trained DM is used as a classifier (“CLF”), then the attack success rate (ASR) for backdoor trigger-injected image prediction remains high. This is similar to the conventional backdoor attacks for image classification.
>
> (**Preserved distribution-level generation ability justified by FID**) Like the basic backdoor attack in image classification, the utility of the BadNets-trained DM is still preserved as it is able to generate high-quality, high-fidelity images. Different from classification, the performance of DM should be evaluated at the data distribution level. As shown in Figure 2 “FID of normal DM v.s. backdoored DM”, the BadNets-trained DM yields a very similar FID score compared to the normally trained DM.
>
> > [R1] Sun, Mingjie, Siddhant Agarwal, and J. Zico Kolter. "Poisoned classifiers are not only backdoored, they are fundamentally broken." arXiv preprint arXiv:2010.09080 (2020).
> >
> > [R2] Wang, Ren, et al. "Practical detection of trojan neural networks: Data-limited and data-free cases." Computer Vision–ECCV 2020: 16th European Conference, Glasgow, UK, August 23–28, 2020, Proceedings, Part XXIII 16. Springer International Publishing, 2020.

---

> ### Author Response · Authors · 2023-11-22
> **Reply to the clarification.**
>
> Q1: Clarificaiton on "Q4"
>
> A1: Thank you for the careful reading and commenting. We agree that input noise distribution is independent from the sampling process in your context and we would like to follow your idea to improve our presentation.
>
> Q2: Clarificaiton on "Q7"
>
> A2: Sorry for the misunderstanding. Please note we mainly visualize the top 100 samples with the highest similarity between A and B  vs. the  similarity between B and C. We thought that the similarity between A and B could be a quantity of greater interest, as it measures the replication level of the generation. This is also supported and presented by Figure 5a in [R1]. However, the magnitude of the similarity score is consistent for measuring (A, B) and (B, C), thus the insights and visualizations provided into (A, B) hold for (B,C). We have added more examples for (B, C) visualization at [**this link**](https://ibb.co/3kqtWLS).
>
> > [R1] Somepalli, G., Singla, V., Goldblum, M., Geiping, J., & Goldstein, T. (2023). Diffusion art or digital forgery? investigating data replication in diffusion models. In Proceedings of the IEEE/CVF Conference on Computer Vision and Pattern Recognition (pp. 6048-6058).

---

### Official Review · Reviewer_8NBW · 2023-10-30

**Soundness:** 3 good
**Presentation:** 3 good
**Contribution:** 3 good
**Rating:** 6
**Confidence:** 3

**Summary:**

The paper inspects the effect of the classic BadNets-like backdoor attacks on diffusion model (DM) training. It discovers bilateral backdoor effects: (1) BadNets-like backdoor attacks cause the trained diffusion models to produce a portion of incorrect images that do not align with the text conditions, and (2) the poisoned diffusion models produce increased ratios of images with the backdoor triggers. Based on the second observation, the paper proposes a novel backdoor defense via training DM models on the suspicious dataset. Datasets with high poisoning rates will produce DMs with amplified trigger appearance, thus allowing more accurate backdoor detection. Datasets with low poisoning rates, instead, have the DMs transform malicious data into benign and be safe to be used for retraining. Backdoored DMs can also be used as image classifiers with improved robustness. Finally, the paper shows that poisoning the replicated training data leads to increased data replication.

**Strengths:**

- The paper shows that the simple, classic BadNets-like backdoor attacks can highly affect the produced DMs. It discovers bilateral backdoor effects, which are somewhat surprising and interesting.
- Based on the observations on BadNets-like backdoored DMs, the paper proposes a novel backdoor defense scheme. It helps improve backdoor detection on datasets with moderate-to-high poisoning rates while producing cleaner data from datasets with low poisoning rates.
- The paper also provides some other insights on the BadNets-like backdoored DMs, which are interesting and potentially useful.

**Weaknesses:**

- As for the first observation, the mismatching between the generated images and the text conditions may not come from backdoor trigger injection but from the incorrect labels of the poisoned images. The authors should try relabeling data only (without adding the backdoor trigger) and check if the ratios of generated mismatching images (G1+G3) are similar to those of BadNets-like backdoored DMs in Figure 3.
- The paper only inspects two simple dirty-label attacks, which are easily detected by standard backdoor defenses. It is recommended to examine if the bilateral effects still appear with more sophisticated dirty-label (e.g., WaNet, LIRA) and clean-label attacks (e.g., LabelConsistent, SIG) and whether the proposed backdoor defense is helpful in those cases.
- The authors should provide more details on the Caltech-15 dataset, including the selected classes, the statistics on data size, and the absolute ratios of poisoned and clean images of the target class.
- In Table 3, the proposed approach only helps to improve backdoor detection accuracy on the generated images; it cannot be used to filter backdoor examples in the original training data. Hence, it does not help to mitigate datasets with moderate-to-high poisoning rates, unlike many other backdoor defense approaches.
- In Table 4, the classifiers trained on generated data, while being more robust, have significant accuracy drops (sometimes more than 10%). Hence, they are not usable in practice. Particularly, since it is hard to differentiate a clean dataset from a dataset with a low poisoning rate, applying this technique will hurt a lot if the suspected dataset is actually clean.
- The test dataset used for the diffusion classifiers (Table 5) is different from the datasets used in the previous experiments. The authors should explain the reason.
- What datasets were used in Table 4, Table 6, and Figures 6-7?

**Questions:**

- The authors should try relabeling data only (without adding the backdoor trigger) and check if the ratios of generated mismatching images (G1+G3) are similar to those of BadNets-like backdoored DMs in Figure 3.
- It is recommended to examine if the bilateral effects still appear with more sophisticated dirty-label (e.g., WaNet, LIRA) and clean-label attacks (e.g., LabelConsistent, SIG) and whether the proposed backdoor defense is helpful in those cases.
- The authors should provide more details on the Caltech-15 dataset, including the selected classes, the statistics on data size, and the absolute ratios of poisoned and clean images of the target class.
- The test dataset used for the diffusion classifiers (Table 5) is different from the datasets used in the previous experiments. The authors should explain the reason.
- What datasets were used in Table 4, Table 6, and Figures 6-7?

---

> ### Author Response · Authors · 2023-11-22
> **Point-to-point Response to Reviewer 8NBW (Part I)**
>
> We genuinely appreciate your diligent review and the thorough summary of our manuscript. Please find our response below.
>
> **Q1**: Question on the generation mismatching may not come from backdoor trigger injection but from the incorrect labels of the poisoned images, relabeling only should be evaluated.
>
> **A1**: We acknowledge the reviewer's concern that image-prompt mismatching may be attributed to incorrect labels instead of trigger patterns. In response, we conducted additional experiments in which the backdoor dataset was constructed by relabeling only. With the generation dissection and composition results presented in [**this link**](https://ibb.co/724wKj6). To ease the comparison with the BadNets attacks, we refer to the generations mismatching the input condition as G1. We observed that “relabeling only” can result in mismatching generations. However, compared to G1 in Fig. 3, the BadNets trigger is absent. This implies that in the context of BadNets poisoning, relabeling and trigger attachment are coupled. Moreover, in Fig. 3, the BadNets poisoning also introduces the G2 type adversarial generations, which can not be achieved by “relabeling only”. As highlighted in the backdoor attack literature for image classification [R1, R2], the basic backdoor attack can be elucidated through the lens of association (a.k.a “shortcut”) between trigger pattern and target label. In the context of image generation, we impose such a **backdoor association** between the target prompt and the image trigger pattern. The created “shortcut” can be revealed by the aforementioned G1 and G2: (a) In the generated images, the proportion of images that mismatch the target prompt and contain triggers (G1) exceeds the corresponding proportion in the training set, as illustrated in Figure 3. (b) The backdoored DM generates trigger-polluted images within the target class that is not poisoned during training (referred to as G2).
>
> Furthermore, our research is not restricted to the adversarial effect of BadNets-type poisoning, but also delves into what insights the backdoored DMs can provide for data replication of DMs and defense against backdoored dataset over DM-generated data. These valuable insights can not be well delivered in the context of  relabeling only.
>
> [R1] Sun, Mingjie, Siddhant Agarwal, and J. Zico Kolter. "Poisoned classifiers are not only backdoored, they are fundamentally broken." arXiv preprint arXiv:2010.09080 (2020).
>
> [R2] Wang, Ren, et al. "Practical detection of trojan neural networks: Data-limited and data-free cases." Computer Vision–ECCV 2020: 16th European Conference, Glasgow, UK, August 23–28, 2020, Proceedings, Part XXIII 16. Springer International Publishing, 2020.
>
> **Q2**: Question on whether the defense insights hold on more sophisticated attacks？
>
> **A2**: We appreciate the suggestion for exploring other sophisticated backdoor methods.
>
> First, we would like to clarify that conducting an in-depth analysis of just the basic backdoor poisoning against DMs has been highly nontrivial. This is our intention to focus on this simplest BadNets-type poisoning setting that has been overlooked in the existing DM literature.
>
> Second, we do value the reviewer’s suggestion and have extended our experiment on the defensive insights in Table 4 to the WaNet attack. We configured the grid size to 224 and set the wrapping strength to 1 to ensure the compatibility of the WaNet attack with ImageNette, which is consistent with previous research [R1].
> All the additional results are presented in [**this link**](https://ibb.co/SsBb0jh). As we can see, the attack success rate (ASR) for an image classifier ResNet-50 trained on the WaNet-backdoored DM-generated dataset is significantly reduced compared to that trained on the original poisoned training set. This defensive insight is consistent with BadNets-type attacks.
>
>
> >[R1] Zhu, Zixuan, et al. "The Victim and The Beneficiary: Exploiting a Poisoned Model to Train a Clean Model on Poisoned Data." Proceedings of the IEEE/CVF International Conference on Computer Vision. 2023.

---

> ### Author Response · Authors · 2023-11-22
> **Point-to-point Response to Reviewer 8NBW (Part II)**
>
> **Q3**: Question on more details on the Caltech-15 dataset.
>
> **A3**: We are very grateful to receive your interest in the details of the Caltech-15 dataset. To construct a subset of Caltech15, we carefully select the 15 categories with the largest sample size from Caltech256. The detailed category names and representative samples for each category are presented at [**this link**](https://ibb.co/WzJZh7h). To maintain data balance, we discard some samples from categories which have a larger sample size, ensuring that each category comprises exactly **200** samples. We designate "**binoculars**" as the target class.
> As for the poisoning ratio with the target class, we randomly select p percent of the samples from non-target categories to inject backdoor. Therefore,  the absolute ratios of poisoned samples within the target class is $\frac{\frac{p}{100}\times N_{nt}}{\frac{p}{100}\times N_{nt}+N_{t}}$, where $N_{nt}$ is the number of non-target class samples and $N_{t}$ denotes the number of target class samples. Please feel free to check our additional experiment results on Caltech15 at [**this link**](https://ibb.co/3fDZMry). Here, the proportion of poisoning training samples within the target class is clearly visualized by the black dashed lines in the sub-figure **e1** and **e2**.
>
> **Q4**: “Weakness” of the proposed backdoor detection method.
>
> **A4**: Thanks for pointing this out. We acknowledge that the backdoored DM cannot be used to filter backdoor examples from the original training data. Instead, our objective is to utilize the backdoored DM as a data transformation operation to discern the presence of backdoor poison at the dataset level, as elaborated on in our manuscript on the last paragraph of Page 7. Moreover, it can be synergized with the latter defense insights which involve training classifiers on the generated dataset. That is, the defenders can firstly detect the presence of the backdoor in the original dataset. If detected, they can then train the classifier on the generated data for improved resilience against data poisoning
>
> The rationale behind our detection insights is the observed “triggered amplification” phenomenon. We view the amplified trigger ratio as an opportunity to enhance the backdoor signature. The left shift in the distribution of detection metrics, presented in [**this link**](https://ibb.co/vzhpfd9), validates the backdoor signature enhancement in the generation phase.
>
> **Q5**: “Weakness” on benign performance degradation of the proposed backdoor defense method.
>
> **A5**: Thank you for raising this question. We admit that  there is a trade-off between accuracy and robustness. However, compared to the decrease in accuracy, the classifier trained on the generated samples shows a more significant decrease in the attack success rate (ASR) than in the accuracy (ACC), as illustrated in Page 8, Table 4. We will add more discussion regarding this tradeoff in the revision.
>
> Meanwhile, regarding the concern about the potential damage to accuracy caused by misapplying this defense to clean dataset. We argue that we can detect first and then decide whether to defend. As illustrated in [**this link**](https://ibb.co/vzhpfd9), the diffusion model can amplify the backdoor signature, making it easier to distinguish between clean data sets and low poisoning rate data sets.
>
>
> **Q6**: Question on what datasets were used in Table 4, Table 6, and Figures 6-7?
>
> **A6**:  In the backdoor defense application (corresponding to Table 4 and Figure 6), we employ both the ImageNette and Caltech15 datasets, as specified in Table 4. For the data replication experiments (corresponding to Table 6 and Figure 7), ImageNette is adopted, as indicated in the second paragraph of Page 9.

---

> ### Author Response · Authors · 2023-11-22
> **Point-to-point Response to Reviewer 8NBW (Part III)**
>
> **Q7**: “Weakness” on the inconsistency between the dataset used for evaluating diffusion classifiers (Table 5) and the datasets used in the previous experiments.
>
> **A7**: Thank you for pointing this out. The main reason for changing the dataset is the considerable computational cost of the stable diffusion classifier, which needs to perform N times diffusion and denoising processes to classify a single sample, where N is typically larger than 100 to achieve high accuracy. Consequently, the time used to classify a sample is N times greater than the time of generating a sample. Even with the timestep sampling strategy and dynamic inference technique proposed in [R1], the stable diffusion classifier requires about 20 seconds to classify a single test sample on a RTX A6000, which limits our ability to conduct large-scale experiments on datasets such as ImageNette and Caltech15. In contrast, CIFAR10 images can be classified faster due to their smaller image size, making the experiments affordable. Please note that another work [R2] leveraging diffusion classifiers for adversarial defense, which also conducts experiments on the CIFAR10 dataset only.
>
> >[R1] Li, Alexander C., et al. "Your diffusion model is secretly a zero-shot classifier." arXiv preprint arXiv:2303.16203 (2023).
> >
> >[R2] Chen, Huanran, et al. "Robust Classification via a Single Diffusion Model." arXiv preprint arXiv:2305.15241 (2023).

---

> ### Comment · Reviewer_8NBW · 2023-12-01
> **The rebuttal addressed my concerns**
>
> Many thanks for the insightful rebuttal. It addressed most of my concerns. I find this paper's main discovery interesting and can benefit the community. The proposed backdoor defenses, however, have some weaknesses limiting their applications. Hence, I decided to keep my initial score.
>
> Best regards,
> Reviewer 8NBW

---

### Official Review · Reviewer_5Vvf · 2023-10-31

**Soundness:** 2 fair
**Presentation:** 3 good
**Contribution:** 2 fair
**Rating:** 5
**Confidence:** 4

**Summary:**

The paper studies the effects of data poisoning in diffusion models (DMs).
Based on the experiments, it finds that backdoored DMs amplify triggers, which
can be used to enhance the detection of backdoor-poisoned training data. The
paper also finds that this can be helpful in designing anti-backdoor
classifiers.

**Strengths:**

Many of the observations made by the paper are very interesting, especially, the
design of anti-backdoor image classifier. I think many of the observations can
be leveraged by future research.

The paper is also easy to follow, and it is well-orgnizaed with claims following
experiments. Sections are also well-connected.

**Weaknesses:**

The paper is rather long and tries to cover too many things, losing the focus.
While many of the claims are interesting. I think the paper limits this
discussion in a narrow aspect, making the results not so convincing. For
example, its claims on poisoning ratios. While existing backdoor attacks have
shown that the poisoning ratio has to be related to many factors -- the training
dataset, the trigger itself, the training objectives and settings, and data
processing techniques (e.g., normalization, smoothing) -- I felt it is too easy
to draw such conclusions base on the limited experiments performed in the paper.
This remains to be the same for **most** claims in the paper, and they make the
paper less convincing to me.

The paper is well-organized, but I would suggest revising the paper a lot. The
abstract dumps many messages, and thought the whole paper, I am not sure what is
the main message of the paper. It seems to be present a set of experiments
(which are not comprehensive), and draw a few conclusions (which seems to be not
convincing based on the description of the experiments). I would suggest
focusing on a single promising direction rather than dumping all experiments.

The paper is very empirical and does not have in depth analysis to explain the
observations, making the observations weak and not strongly supported. I would
wonder if there exists adaptive methods that compromises the observations.

**Questions:**

What is the main takeaway message?

---

> ### Author Response · Authors · 2023-11-22
> **Point-to-point Response to Reviewer 5Vvf (Part I)**
>
> **Q1**: “Weakness” on losing focus.
>
> **A1**: Thank you for acknowledging the claims in our paper. It is true that we put lots of new insights on BadNets-poisoned dataset training for diffusion models in our paper. However, we respectfully disagree that “our paper is losing focus”.
>
> As illustrated in GR1, we maintain a consistent focus on examining the impact of the basic BadNets-like backdoor poisoning on diffusion models (DMs). That is, we aim to understand the consequences when a DM encounters a poisoned dataset reminiscent of BadNets attacks. Previous research has already demonstrated the vulnerability of DMs to backdoor attacks, but these methods typically involve modifying the noise distribution or the training objective, assuming the adversary can not only poison the dataset but also control the model training. This is different from the classical backdoor poisoning techniques that merely involve contaminating the training dataset and do not rely on such additional “training assumptions”. Based on that, we are inspired to investigate a more covert backdoor scenario involving DMs, where the manipulation is confined to the training data alone.
>
> The primary message conveyed by our results revolves around the **bilateral backdoor poisoning effects**, encompassing both the attack perspective, referred to as "Trojan Horses", and the backdoor defense perspective, known as "Castle Walls". A brief overview of these concepts can be found in the second paragraph of the Introduction. These insights are then systematically detailed in Sections 4 and 5, respectively. Furthermore, we establish a connection between this backdoor effect and the data replication effect in DMs (Section 6). To enhance clarity, we have provided concise summaries of the main insights at the outset of each section. As we can see from the summaries, all three aspects and their corresponding subsections within each aspect are organized to synergistically complement one another rather than being treated as separate entities.
>
> **Q2**: “Weakness” on the lack of experiments considering more factors.
>
> **A2**: Thank you for bringing up this question. Your suggestion has inspired us to consolidate and summarize the various factors involved in our experimental setup for data poisoning on the diffusion model. (a) Regarding the training set: we covered CIFAR-10, ImageNette, and Caltech-15 (See additional results at [**this link**](https://ibb.co/3fDZMry)). (b) Regarding training objective: We focused on the basic BadNets setting. Thus the original model training objective is kept intact for backdoor data poisoning. (c) Regarding triggers: We consider the most representative triggers (Table 2) in the BadNets setting used by [R1, R2]. In addition, we have expanded our experimentation to encompass a more sophisticated trigger, called WaNet in [R3]. Our defense insights have been justified under this new trigger (See results at [**this link**](https://ibb.co/SsBb0jh)).
>
> Additionally, in our original submission, we have also considered two other crucial factors: the data poisoning ratio and the guidance weight applied to DM training (see Fig. 4). We would like to draw attention to these factors and underscore our commitment to comprehensively exploring their impact in the context of data poisoning.
>
>
> >[R1] Chen, X., Liu, C., Li, B., Lu, K., & Song, D. (2017). Targeted backdoor attacks on deep learning systems using data poisoning. arXiv preprint arXiv:1712.05526.
> >
> >[R2] Gu, Tianyu, et al. "Badnets: Evaluating backdooring attacks on deep neural networks." IEEE Access 7 (2019): 47230-47244.
> >
> >[R3] Nguyen, A., & Tran, A. (2021). Wanet--imperceptible warping-based backdoor attack. arXiv preprint arXiv:2102.10369.

---

> ### Author Response · Authors · 2023-11-22
> **Point-to-point Response to Reviewer 5Vvf (Part II)**
>
> **Q3**: “Weakness” on the lack of in-depth analysis and explanation.
>
> **A3**: We respectfully disagree with the assessment that our analysis and explanations lack depth. Our paper is structured to convey clear and comprehensive insights into the “bilateral” backdoor poisoning effects we have identified. We begin by organizing our findings at the highest levels, under the headings of the bilateral effects "Trojan Horses" and "Castle Walls," and proceed to dissect each of these concepts into various perspectives. For instance, when delving into the "Trojan Horses" aspect, we provide an intricate analysis by breaking down the compositions of generated images. This allows us to offer insights into trigger amplification and phase transition. Subsequently, as discussed in the "Castle Walls" section, we derive two applications from these analyses: 1) backdoor poisoned dataset detection, inspired by the observations related to trigger amplification, and 2) a backdoor defense strategy employing synthetic data training and diffusion classifiers, inspired by the insights surrounding the phase transition. Moreover, we establish a meaningful connection between the observed backdoor effect and the data replication effect in DMs, as elaborated upon in Section 6. In sum, our paper strives to provide a thorough and interconnected exploration of these concepts.
>
> **Q4**: Question on the main takeaway.
>
> **A4**: As explained in the responses A1 and A3, the main takeaway is the **bilateral backdoor poisoning effects**, encompassing both the attack perspective, referred to as "Trojan Horses", and the backdoor defense perspective, known as "Castle Walls". To enhance clarity, we have provided concise summaries of the main insights at the outset of each technical section.

---

### Official Review · Reviewer_UGJ5 · 2023-11-01

**Soundness:** 4 excellent
**Presentation:** 3 good
**Contribution:** 3 good
**Rating:** 6
**Confidence:** 4

**Summary:**

Since previous works show that diffusion models are still threatened by backdoor attacks, this paper investigates whether backdoor attacks for diffusion models can be as simple as "BadNets". After directly adding poison images into training datasets and substitute the benign prompt with the misaligned prompt. The answer to the above question is "yes". In addition to this empirical finding, this paper also provide a mount of fruitful insights: The phase transition phenomenon, the trigger amplification effect and correlation with data replications. Overall, this paper is a quite solid work.

**Strengths:**

1 This paper is well-written.

2 This paper is easy to follow.

3 This paper provides multiple insights which are rarely mentioned in previous works.

**Weaknesses:**

1 minor errors:

​       (1) Page 5 should be "Table 2" instead of Figure 2.

2 All experiments are performed on small dataset. Do the provided insights still hold on larger datasets, such as ImegNet-100?

**Questions:**

1 Can you explain why in Figure 2, poisoning 10% of training samples will have marginal effect on the FID of benign images? Does this mean that the performance of diffusion model is not sensitive to the number of training dataset?

2 ddim is another sampling algorithm for diffusion models. Would your proposed insight vary if we use DDIM to sample images from a backdoor diffusion models?

3 Can your provided insights be transferable to SDE [1]?

[1] Yang Song, Jascha Sohl-Dickstein, Diederik P. Kingma, Abhishek Kumar, Stefano Ermon, and Ben Poole.

---

> ### Comment · Reviewer_UGJ5 · 2023-11-22
> **Look forward to your rebuttal**
>
> I think it is a quite good paper. As a reviewer, I am looking forward to receiving rebuttals from the authors.

---

> > ### Author Response · Authors · 2023-11-22
> > **Thank you for the encouragement!**
> >
> > Dear Reviewer UGJ5,
> >
> > We sincerely appreciate your words of encouragement. We are currently finalizing our rebuttal and anticipate releasing it within the next 24 hours.
> >
> > Thank you very much,

---

> ### Author Response · Authors · 2023-11-22
> **Point-to-point Response to Reviewer UGJ5 (Part I)**
>
> **Q1**: "Table 2" instead of Figure 2？
>
> **A1**: Thanks for pointing this out, we will correct it.
>
> **Q2**: Question on whether our insights still hold on larger datasets like ImageNet-100?
>
> **A2**: Thanks for raising the question. Inspired by that, we have extended our experiments to a new dataset **Caltech15**, a subset comprising 15 categories from Caltech256. The obtained results on Caltech15 can be accessed through [**this link**](https://ibb.co/3fDZMry). Notably, the effect of backdoor attacks on the Caltech15 dataset is more pronounced than on ImageNette. For instance, when employing a poisoning ratio of 5% and a guidance weight of 5, the proportion of trigger-presented images (G1+G2) generated by the backdoored DM is 47.5% higher than the proportion of the poisoned training set. In contrast, on ImageNette, the increase in trigger proportion is 17.2%. Our experiments on both ImageNette and Caltech15 affirm that the BadNets-like dataset remains effective to backdoor DMs, causing incorrect (G1) or tainted (G2) image generation.
>
> Unfortunately, we were not able to extend our experiments on ImageNet-100 during the rebuttal phase due to our currently limited computing resources. Please note that it takes much more intensive computational costs for training backdoor-poisoned diffusion models than classifiers. We will continue conducting this experiment by requesting more computing resources after rebuttal. Meanwhile, it is worth noting that prior works [R1,R2] on backdoor diffusion models have also mainly conducted experiments on CIFAR10 and CelebA. Thus, we believe that the insights we gained should still hold on larger datasets.
>
> > [R1] Chou, Sheng-Yen, Pin-Yu Chen, and Tsung-Yi Ho. "How to backdoor diffusion models?." Proceedings of the IEEE/CVF Conference on Computer Vision and Pattern Recognition. 2023.
> >
> > [R2] Chen, Weixin, Dawn Song, and Bo Li. "Trojdiff: Trojan attacks on diffusion models with diverse targets." Proceedings of the IEEE/CVF Conference on Computer Vision and Pattern Recognition. 2023.
>
>
> **Q3**: Question on why poisoning 10% of training samples will have marginal effect on the FID of benign images?
>
> **A3**: Thank you for raising this question. Poisoning 10% of the training samples does have marginal effect on the quality of the benign samples, as assessed by the Fréchet Inception Distance (FID) score between the generated images of non-target classes and the training images of non-target classes (see the second paragraph at Page 5). We observe that generated images under non-target classes typically do not contain triggers, and can thus be well aligned with the non-target class data in the training set. This observation bears a resemblance to the phenomenon of a backdoored classifier achieving high classification accuracy on clean data.
>
> As for whether the performance of the DM is insensitive to the dataset size, we believe that reducing a *moderate proportion* of training data will not significantly degrade the DM’s performance. This assertion is supported by previous research on backdoor DMs. For example, [R1] has shown that a backdoored DM sustains performance comparable to its normally trained counterpart, even when subjected to larger poisoning ratios ranging from 10% to 30%. In fact, the reviewer's question has inspired us to explore an important aspect: dataset pruning for diffusion models. This line of inquiry seeks to understand how the training set can be selectively pruned without hampering the performance of DMs.
>
> > [R1] Chou, Sheng-Yen, Pin-Yu Chen, and Tsung-Yi Ho. "How to backdoor diffusion models?." Proceedings of the IEEE/CVF Conference on Computer Vision and Pattern Recognition. 2023.
>
>
> **Q4**: (Question on extending to DDIM samplers) DDIM is another sampling algorithm for diffusion models. Would your proposed insight vary if we use DDIM to sample images from backdoor diffusion models?
>
> **A4**: Our insights hold on DDIM. In fact, all of our results on stable diffusion were obtained using DDIM sampler with sampling steps = 50 and eta = 0. DDPM is only employed for generation on CIFAR-10 using a relatively small diffusion model, as we have pointed out in the first paragraph of Page 5.

---

> ### Author Response · Authors · 2023-11-22
> **Point-to-point Response to Reviewer UGJ5 (Part II)**
>
> **Q5**: Question on whether our insights hold on SDE sampler?
>
> **A5**: Yes, the backdoor threat also exists on the SDE sampling [R1] and the obtained results are presented in [**this link**](https://ibb.co/jw571DQ). However, we observed that the backdoored DM generates less trigger-tainted images using SDE sampling. We attribute this observation to the increased randomness introduced by SDE sampling [R2], consequently hindering the replication of trigger patterns.
>
> > [R1] Song, Y., Sohl-Dickstein, J., Kingma, D. P., Kumar, A., Ermon, S., & Poole, B. (2020). Score-based generative modeling through stochastic differential equations. arXiv preprint arXiv:2011.13456.
> >
> >[R2] Lu, C., Zhou, Y., Bao, F., Chen, J., Li, C., & Zhu, J. (2022). Dpm-solver++: Fast solver for guided sampling of diffusion probabilistic models. arXiv preprint arXiv:2211.01095.

---

### Author Response · Authors · 2023-11-22
**Highlighted General Response (GR)**

**GR1**: **Possible misunderstanding of the goals of our work**

A few concerns were raised by reviewers about the goal (@Reviewer TC33) and the focus (@Review 5Vvf) of our work. We would like to address potential misconceptions about our paper and provide a clear clarification regarding the motivation and logic behind it.

In our study, we maintain a consistent focus on examining the impact of the basic BadNets-like backdoor poisoning on diffusion models (DMs). That is, we aim to understand the consequences when a DM encounters a poisoned dataset reminiscent of BadNets attacks. Previous research has already demonstrated the vulnerability of DMs to backdoor attacks, but these methods typically involve modifying the random noise distribution or the training objective, assuming the adversary can not only poison the dataset but also control the model training. It is crucial to emphasize that classical backdoor poisoning techniques merely involve contaminating the training dataset and do not rely on such additional “training assumptions”. This realization has prompted us to investigate a more covert backdoor scenario involving DMs, where the manipulation is confined to the training data alone.

By introducing poisoned data into the training set using a basic BadNets-like approach, we have observed two intriguing “attack phenomena”, which we have termed "trigger amplification" and "phase transition." These observations have led us to propose valuable insights concerning backdoor poisoning attack and defense, which we have subsequently validated through experimental analysis. To clarify, the backdoor defense insights presented in our paper are not intended for detecting backdoors within diffusion models themselves but rather leverage DMs as a transformative tool to ascertain whether suspicious training data has been compromised.

**GR2**: **A summary of additional experiments**

We have made a substantial effort to enrich our experiments based on reviewers’ suggestions (see the revised PDF). Below is a summary, where Q-$i$ represents the $i$-th question in our response:

- **Reviewer** UGJ5
    - **Q2**: Experiments on Caltech-15.
    - **Q5**: Experiments using SDE sampler.
- **Reviewer** 5Vvf
    - **Q2**: Experiments on other dataset and considering more factors.
- **Reviewer** 8NBW
    - **Q1**: Experiments with the “relabel only” attack.
    - **Q2**: Experiments with more sophisticated attacks.
    - **Q3**: Experimental details on Caltech-15 dataset.
    - **Q4&Q5**: Experiments on distribution shift of backdoor signatures.
- **Reviewer** TC33
    - **Q6**: Experiments on distribution shift of backdoor signatures.
    - **Q7**: More visualization over replicated data.

---

### Meta-Review · Area_Chair_VmYf · 2023-12-05

**Metareview:**

Based on reviewers' viewpoint, the authors partially address their comments,
In particular, Reviewer TC33's concern is that a well-defined backdoor should be precise and stealthy (as mentioned by the second point in the weaknesses).  From his/her perspective, the authors' responses didn't address this concern (ie, the precise and stealthy requirements).

**Justification For Why Not Higher Score:**

Reviewers' concern are not entirely addressed.

**Justification For Why Not Lower Score:**

The authors' responses satisfy at least two reviewers.

---

### Decision · Program_Chairs · 2024-01-16

Reject